# A Study on the Relationship between Road Network Centrality and the Spatial Distribution of Commercial Facilities—A Case of Changchun, China

Xiaochi Shi [1], Daqian Liu [2,*] and Jing Gan [1]

1 School of Geographical Sciences and Tourism, Jilin Normal University, Siping 136000, China; shixiaochi@iga.ac.cn (X.S.); ganjing@jlnu.edu.cn (J.G.)
2 State Key Laboratory of Black Soils Conservation and Utilization, Northeast Institute of Geography and Agroecology, Chinese Academy of Sciences, Changchun 130102, China
* Correspondence: liudaqian@iga.ac.cn

**Abstract:** Using the Urban Network Analysis Tool, the centrality of a road network (closeness centrality, betweenness centrality, and straightness centrality) was calculated, and the POI data of the commercial facilities were reclassified. KDE estimation was used to estimate the centrality of the traffic network, and the correlation coefficient was calculated to explore the spatial relationship between road network centrality and the types of commercial facilities (catering facilities, shopping facilities, residential life facilities, and financial and insurance facilities). The results indicate the following: (1) Closeness centrality displays a discernible "Core–Periphery" pattern, and the high-value areas of betweenness centrality are mainly concentrated around the main arterial roads of the city. In contrast, straightness centrality unveils a polycentric structure. (2) The spatial distribution of commercial facilities demonstrates a notable correlation with the centrality of the road network. From the perspective of centrality, the distribution of residential life facilities is most strongly influenced by road network centrality, followed by financial and insurance facilities and then catering facilities, with the distribution of shopping facilities being the least affected. (3) The centrality of the road network plays a crucial role in shaping the arrangement of commercial facilities. Closeness centrality significantly influences the distribution of residential life facilities, catering facilities, and shopping facilities. Betweenness centrality has a noteworthy impact on the selection of locations for financial and insurance facilities, as well as residential life facilities. Furthermore, areas characterized by better straightness centrality are preferred for the distribution of residential life facilities, financial and insurance facilities, and catering facilities. (4) The centrality of the road network has a greater influence on the arrangement of various commercial facilities than the population distribution.

**Keywords:** road network centrality; multi-center evaluation model; commercial facilities; Changchun main urban area

## 1. Introduction

Network methodologies have long been integral to economic geography and urban planning. As transportation networks develop and improve, regional accessibility is consistently enhanced, which, in turn, facilitates the spatial movement of various productive forces. The evolution of urban road network characteristics has stimulated urban economic activities, leading to notable variations in urban functional land use [1]. Concurrently, urban functional land use influences the travel behaviors of residents [2–4]. Recent research on urban road networks has employed various methodologies. Notably, scholars have utilized the Analytic Hierarchy Process (AHP) to classify roads based on their importance and to analyze optimal routes, thus enhancing the quality of road services [5–7]. Other scholars have segmented roads into Traffic Analysis Zones (TAZs), constructed origin–destination (OD) matrix formats by pinpointing origins and destinations, and optimized

the traffic network structure through distance calculations [8–10]. Additionally, Mahmoud Owais [11,12] and colleagues have advanced the OD model to enhance public transit networks by assessing the number of route-covered node pairs and the Passenger Transfer Number (PTN), thereby improving the connectivity and directness of public transportation. The Analytic Hierarchy Process (AHP) ranks roads based on specific criteria and does not consider all levels of roads as a unified system. Furthermore, OD analysis strongly relies on the model, and the model's development may not adequately capture the full complexity of the road network. Centrality analysis, however, treats the entire road network as a cohesive system for analysis, where roads are viewed as edges and intersections as nodes to assess the significance or influence of a node within the network. Centrality, a core concept in network analysis science, measures the importance of nodes within a network and is vital for characterizing the features of urban road networks [13,14]. Initially emerging from structural sociology [15,16], the concept of centrality has been applied to examine geographical relationships among traffic flow, human movement, financial transactions, logistics, and land use [17].

The multi-center evaluation model (MCA) is commonly employed to quantify the centrality of spatial networks [18]. Scholars have utilized the multi-centrality evaluation model to quantitatively assess the structural characteristics of road networks. Sergio et al. examined the accessibility of the road network system [18], while Miller evaluated the spatiotemporal accessibility of the road network [19]. Furthermore, scholars have utilized the multi-centrality evaluation model to comprehensively quantify the global centrality characteristics of road networks, including measures such as betweenness centrality [20], closeness centrality [21], straightness centrality, and eigenvector centrality [22]. These metrics are analyzed in relation to economic activities [23–27], population density [28], land use [29–33], the layout of functional land uses [34,35], urban dynamic processes [36–38], and emerging trends in urban structure [39]. Additionally, some researchers have employed road network centrality models to examine the degree of connectivity in transportation roads [39,40], the complementarity of traffic flows [41], and their impact on residential rental prices [42]. Consequently, the relationship between the distribution of urban economic activities and the centrality of transportation networks has garnered widespread attention across diverse academic disciplines. The aforementioned research findings rely on MCA indices, including closeness centrality, betweenness centrality, and straightness centrality, to assess the centrality of transportation networks. These indices are crucial in fields such as urban and rural planning, land use, and the layout of urban spatial structures in various countries. Previous research has achieved notable results, primarily focusing on the relationship between road network centrality and economic development, as well as its association with land use. However, to our knowledge, there is a scarcity of studies examining the impact of road network centrality indices on the spatial distribution of various types of commercial facilities, including catering facilities, shopping facilities, residential life facilities, and financial insurance facilities. Investigating the influence of commercial service facilities on the structure of road networks is crucial for leveraging the road transportation system to enhance the spatial development of commercial facilities.

The street network of roads is the skeleton of the city [43]. Multi-center evaluation models have become increasingly popular as a methodology for studying urban road systems. Individuals reside, work, and engage in recreational activities within a network interconnected by roads spanning various locations. The road traffic network serves as a vital link between the spatial configuration of the city and the distribution of urban economic activities. The city's spatial structure often exhibits concentric circles, sectors, and multi-center arrangements, with the centered traffic network's design playing a pivotal role in shaping the city's spatial layout. The configuration of the traffic network determines the spatial structure of the city, thereby influencing the intricate spatial arrangement of Points of Interest (POIs). In this study, we computed the closeness centrality, betweenness centrality, and straightness centrality of the road network using the Urban Network Analysis Tool (UNA), employing the city center of Changchun as an illustrative example within the

framework of the multi-center evaluation model (MCA). Subsequently, we conducted an analysis of the spatial distribution of commercial facilities by integrating Point of Interest (POI) data. Subsequently, in conjunction with Point of Interest (POI) data, we analyzed the spatial distribution of commercial facilities. Utilizing KDE estimation, we standardized both components to a common spatial unit, enabling an assessment of the relationship between the former and the latter. This research is dedicated to unveiling the underlying causal mechanisms linking transportation network centrality and commercial facilities. Its objective is to establish a theoretical foundation for optimizing the city's internal structure, improving commercial facility layouts, and augmenting their economic benefits.

## 2. Study Area and Data

### 2.1. Study Area

Changchun, the capital of Jilin Province, is situated in the northeastern part of China and exemplifies a single-center pattern as an inland city. The city encompasses a significant number of roundabouts within its boundaries. It features a "two horizontal and three vertical" expressway system alongside a "two horizontal, one vertical, two ring, and four radiating" main road network, creating a grid-like and three-dimensional transportation system. This road network significantly influences the distribution of commercial facilities. Additionally, Changchun is also known as the "Spring City of the North". Serving as the core city of the Harbin–Changchun Urban Agglomeration, it stands as a crucial industrial base and serves as the pilot city for the "Made in China 2025" initiative. In 2022, Changchun had a population of 9.065 million permanent residents and a gross domestic product (GDP) of approximately USD 942.2 billion.

In this study, our research is centered on the primary urban area of Changchun City, specifically encompassing Nanguan District, Chaoyang District, Lvyuan District, Kuancheng District, and Erdao District, covering an area of approximately 612 square kilometers. This region serves as the political, economic, and cultural nucleus of the province. Over time, substantial infrastructure development has occurred within the main urban area of Changchun, which has led to the establishment of an evolving transportation system. The total length of roads in the main urban area is approximately 1546.1 km, consisting mainly of trunk roads, expressways, bypasses, and secondary trunk roads (see Figure 1).

### 2.2. Data

The road network data used in this study were derived from the 2020 edition of the urban road system planning map. The main urban area road network of Changchun City was spatially registered and vectorized at various levels. By overlaying roads of various grades, a road network system was constructed, with road intersections and endpoints serving as network nodes, resulting in a total of 3918 extracted nodes. Point of Interest (POI) data, comprising commercial facility points (including catering facilities, shopping facilities, residential life facilities, and financial and insurance facilities) for the year 2022, were obtained through the Gaode Map API, amounting to a total of 118,458 data points. The detailed categorization of these POIs is presented in Table 1.

**Table 1.** Types and numbers of POIs served by commercial facilities in Changchun's main urban area.

| Category 1 | Category 2 | Category 3 | Number of POIs |
|---|---|---|---|
| | catering facilities | Fast food, dessert store, coffee shop, restaurant, etc. | 28,584 |
| Commercial facilities | shopping facilities | Convenience stores, markets, supermarkets, etc. | 34,486 |
| | residential life facilities | Public restrooms, logistics, beauty salons, business offices, etc. | 52,964 |
| | financial and insurance facilities | ATMs, banks, financial institutions, etc. | 2424 |

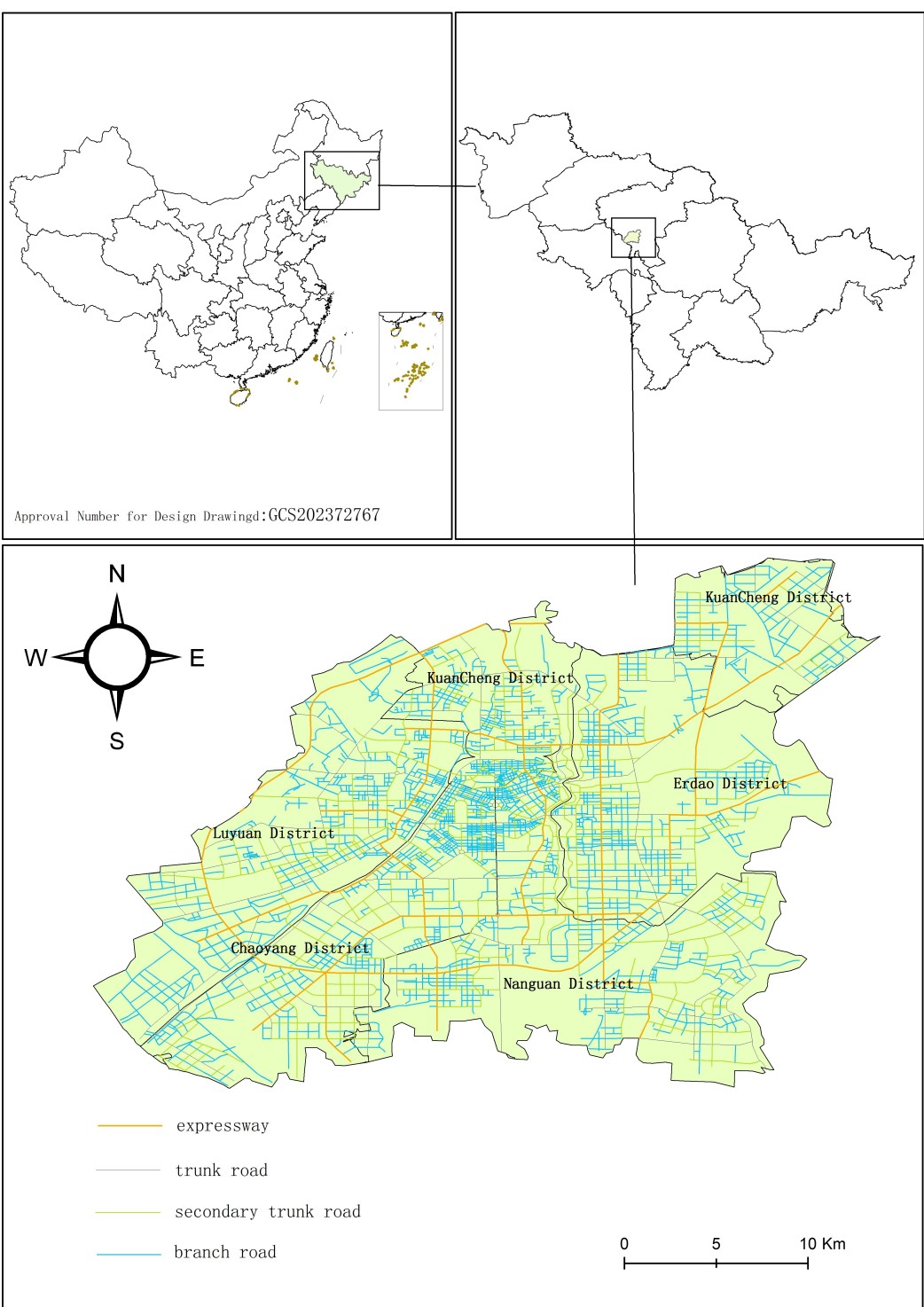

**Figure 1.** The main urban area of Changchun City. Note: The map of China is from the standard map service system: http://bzdt.ch.mnr.gov.cn (accessed on 4 August 2023).

## 3. Methods

This paper calculates the centrality of the road network using a multi-center evaluation model and transforms the centrality values, along with the kernel density values of various types of commercial facilities and the population distribution, into a unified raster system through kernel density estimation methods. This standardization facilitates the assessment of their relationships on a consistent scale.

### 3.1. Multi-Center Evaluation Model

The multi-center evaluation model is a fundamental analytical approach used in urban design and planning processes [44]. This paper employs the multi-center evaluation model to study the centrality of road networks within a specified scope. The core concept involves abstracting urban roads as edges of a network, with intersections or endpoints of two or more roads acting as nodes. By measuring the distances between nodes along the actual network paths, the centrality of the transportation network is calculated [24]. In the multi-center evaluation model, three indicators are particularly important [18]: closeness centrality, betweenness centrality, and straightness centrality. These metrics assess whether a location is proximate to all other locations, serves as an intermediate point for other locations, and can be directly reached from all other points, respectively.

#### 3.1.1. Closeness Centrality

Closeness centrality serves as a measure that indicates how proximate a node is to all other nodes within the network. It is defined as the reciprocal of the average shortest path length from the node to all other nodes in the network, excluding the node itself. Closeness centrality, being inversely proportional to the average distance, emphasizes the pattern of distance attenuation within the urban road network. The calculation formula is as follows:

$$C_i^C = \frac{N-1}{\sum_{j \neq k} n_{jk}} \tag{1}$$

$C_i^C$ is the closeness centrality of nodes; N is the total number of nodes in the transportation network; and $n_{jk}$ is the shortest distance from node j to point k.

#### 3.1.2. Betweenness Centrality

Betweenness centrality quantifies the number of shortest paths passing through a specific node in a network. A higher betweenness centrality indicates that a node is more frequently positioned on the shortest path between pairs of other nodes. In the context of road networks, a road with high betweenness centrality is more likely to be selected for travel, serving as a crucial intermediate point rather than an origin or destination. Betweenness centrality can effectively indicate the magnitude of traffic flow at road network nodes [26]. The calculation formula is as follows:

$$C_i^B = \frac{1}{(N-1)(N-2)} \sum_{j=1;k=1;j \neq k \neq 1}^{N} \frac{d_{jk(i)}}{d_{jk}} \tag{2}$$

$C_i^B$ is the betweenness centrality of nodes; N is the total number of nodes in the transportation network; $d_{jk}$ is the number of shortest paths from node j to node k; and $d_{jk(i)}$ is the number of shortest distances passing through node i between node j and node k.

#### 3.1.3. Straightness Centrality

Straightness centrality is defined as the average ratio of the shortest distance between two nodes in a network to the straight-line distance between them. This metric provides insight into the significance of a node within the network. The higher the straightness centrality, the closer the shortest path between two nodes to their straight-line distance, indicating better traffic efficiency for the network nodes. Straightness centrality stands as a crucial index for assessing the access efficiency of transportation networks [26]. The calculation formula is as follows:

$$C_i^S = \frac{1}{N-1} \sum_{j=1;j \neq i}^{N} \frac{d_{ij}^{Euel}}{d_{ij}} \tag{3}$$

$C_i^S$ is the straightness centrality of the nodes; N is the total number of nodes in the transportation network; $d_{ij}^{Euel}$ is the Euclidean distance from node i to j; and $d_{ij}$ is the shortest distance from node i to j.

### 3.2. Kernel Density Estimation

Kernel density estimation utilizes the range or window of each cell to represent the density within that area. It relies on a function that assigns greater weights to nearby objects than to distant ones, calculating them using the Euclidean distance [45]. This method measures the properties of a central location by evaluating the density of surrounding objects, thereby capturing the attribute of the location. The attribute of the location is not interpreted in isolation; instead, its significance is derived from its relationship with the surrounding environment [29]. The kernel density estimation method is widely utilized in urban agglomeration analysis. It assesses the dispersion of each sample point within a specified range from the surrounding area, fits observed data points with a peak function, and approximates the distribution of these points [35]. Thus, employing kernel density estimation not only leverages the intrinsic tools of ArcGIS but also enables the precise analysis of the relationship between adjacent features. Furthermore, road network centrality and commercial facilities' distribution represent two distinctly different spatial characteristics. To examine the relationship between urban transport network centrality and the distribution of various types of Points of Interest (POIs), the kernel density estimation method [23–25] is utilized. This method transforms two different spatial elements into a unified space, facilitating the analysis of their relationship.

There are three additional reasons for using kernel density estimation in this article, as described below.

Kernel density estimation (KDE) is a method used to represent the characteristic properties of a central location based on the density of nearby objects, thereby reflecting both centrality and the density of service facilities. KDE focuses not on the place itself but on how its environment distinguishes it and explains its attributes. The use of KDE is not for transforming the format of data but rather for more accurately capturing the true interactions between two adjacent features.

The use of kernel functions in kernel density estimation recognizes the distance-decay property of spatial interactions by emphasizing the contribution of nearby objects over those farther away. This concept is widely acknowledged in urban research fields. Similarly, the gravity model, which follows the same principle, has a strong theoretical foundation and numerous successful applications in urban and regional studies [46].

Kernel density estimation is a standard tool within the spatial analysis toolkit of ArcGIS 10.4, and the results are more easily mapped and visualized using this software.

The calculation formula is as follows:

$$f(x) = \frac{1}{nh} \sum_{i=1}^{n} k\left(\frac{x - x_i}{h}\right) \tag{4}$$

k( ) is the kernel density equation; h is the width; and n is the number of points within the width. In order to highlight the spatial clustering characteristics of different types of POIs and road network centrality in Changchun City and to deeply analyze the relationship between the two, the clustering effect is most obvious when the search radius is set at 1 km and the grid size is set at 50 m ∗ 50 m as the uniform width in the kernel density estimation of the road network centrality and POIs through several experiments.

### 3.3. Pearson Correlation Analysis

The Pearson correlation coefficient measures the degree of linear correlation between the independent variable X and the dependent variable Y, with r ranging in size from −1 to 1. A negative value of r denotes a negative correlation, while a positive value denotes a

positive correlation. A larger absolute value of r indicates a stronger correlation, and if r is 0, there is no correlation. The calculation formula is as follows:

$$r = \frac{\sum_{i=1}^{n}\left(X_i - \overline{X}\right)\left(Y_i - \overline{Y}\right)}{\sqrt{\sum_{i=1}^{n}\left(X_i - X\right)^2}\sqrt{\sum_{i=1}^{n}\left(Y_i - Y\right)^2}} \tag{5}$$

r is the correlation coefficient, and X and Y are the variables.

### 3.4. Multiple Linear Regression

The geo-environmental system is influenced by multiple elements, and there are interactions and interconnections between multiple (two or more) elements, making multiple linear regression models more general. Let the dependent variable Y be affected by the independent variables $(X_1, X_2, X_3 \ldots, X_k)$ with n sets of observations $(Y_a, X_{1a}, X_{2a} \ldots, X_{ka})$, a = 1, 2, ..., n. The multiple linear regression model is calculated as follows:

$$Y_a = \beta_0 + \beta_1 X_{1a} + \beta_2 X_{2a} + \cdots + \beta_k X_{ka} + \varepsilon_a \tag{6}$$

The geo-environmental system is influenced by multiple elements, and $\beta_0, \beta_1, \beta_2 \ldots \beta_k$ are parameters to be determined; $\varepsilon_a$ is a random variable.

## 4. Spatial Distribution Characteristics of Road Network Centrality and Commercial Facilities

### 4.1. Characteristics of Road Network Centrality Distribution

ArcGIS software was employed to establish a road network dataset, which was used to extract the endpoints and intersections of roads, serving as nodes within the traffic network. Subsequently, the Urban Network Analysis Tool (UNAT) was utilized to compute the closeness centrality, betweenness centrality, and straightness centrality of these road network nodes within the primary urban area of Changchun City. Using a search radius of 1 km, the kernel density estimation (KDE) method was applied to determine the centrality kernel density values of the road network (see Figure 2).

Closeness centrality demonstrates a distinctive "Core–Periphery" pattern, consistent with the distance-decay principle in its spatial distribution. On the kernel density distribution map, which reveals a polycentric structure, notable concentrations of high closeness centrality values are observed in Changchun's main urban area, particularly around People's Square, the train station, and major roads such as Renmin Avenue, Shengli Avenue, and Chongqing Road. The traffic network nodes near these roads exhibit the smallest average distance to all network nodes in the surrounding area. The next-highest values predominantly occur within the region bordered by Guilin Road (CBD) to the west. Additionally, areas like Changchun Street, Changtong Road, Jiefang Road South Hutong, Hunchun Street, Tonghua Road, Shuguang Road, and other subsidiary roads near this region also demonstrate elevated closeness values, with smaller average distances from regional transportation network nodes.

The spatial distribution characteristics of betweenness centrality exhibit notable variations when compared to those of closeness centrality. High betweenness centrality values are predominantly concentrated in the northern section of Renmin Avenue and Changchun Avenue. Furthermore, areas around the intersections of Nanku Avenue and Renmin Avenue, the Southern Expressway and Renmin Avenue, and Donghuancheng Road and the Jilin Expressway also exhibit elevated betweenness centrality. This region's transportation network, characterized by high betweenness centrality, indicates the presence of numerous shortest paths passing through this area. The aforementioned areas exhibit higher traffic network intermediacy, with a considerable number of shortest paths within the region, frequent selection for urban travel, substantial traffic flow, and a pivotal role in enhancing the internal urban traffic network.

The spatial distribution characteristics of straightness centrality exhibit similarities to those of closeness centrality, displaying a polycentric structure. High-value areas are

predominantly situated near People's Square and the railway station. They are also found in the region defined by the northern section of People's Street, Changchun Street, and Yatai Street, as well as in the area formed by Xinmin Street, Freedom Street, People's Street, and Jiefang Street. Furthermore, high straightness centrality is observed in the area encompassing Xigangdaogang Street and Hongqi Street. Additionally, the vicinity of Xi'an Hutong, Jiansheng Street, Xichaoyang South Hutong, Hwagwang Lu, and Huichun Street East Hutong also shows elevated straightness centrality. Within these areas, the distance between two nodes in the road network closely approximates the straight-line distance between them. As a result, the network nodes exhibit high straightness centrality, leading to efficient access.

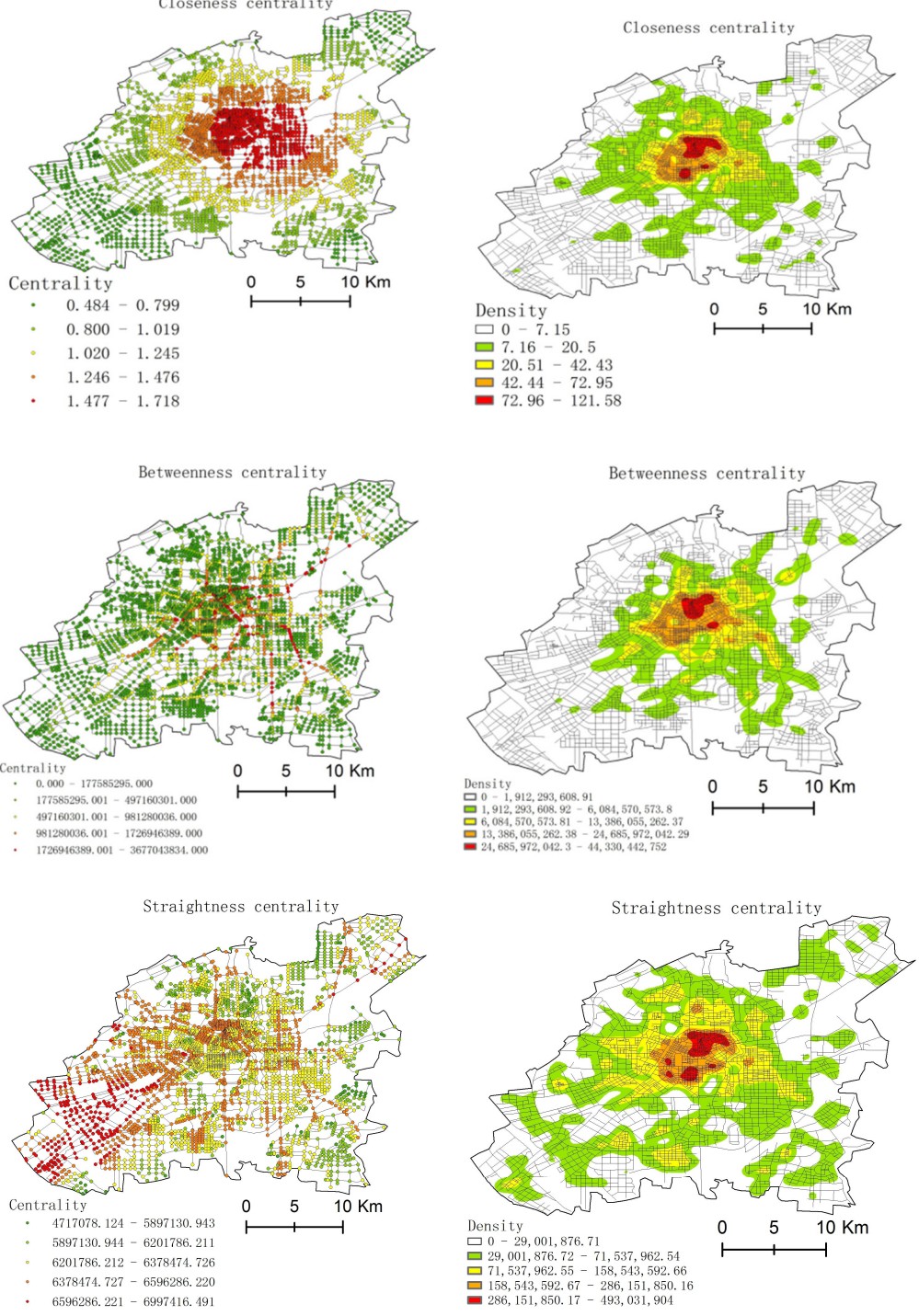

**Figure 2.** Road network centrality and its kernel density distribution.

## 4.2. Distributional Characteristics of Commercial Facilities

To illustrate the spatial distribution of reclassified commercial facility data with a 1 km search radius, we calculated the spatial distribution characteristics of commercial facilities in the main urban area of Changchun City (see Figure 3). In a broad sense, the distribution of commercial facilities is generally characterized by a dense concentration in the west and a sparse distribution in the east, with the Yitong River serving as the dividing line. Delving into specifics, catering and residential life facilities predominantly cluster within the region bounded by Shanghai Road to the east and Dama Road to the west. This corridor extends from Changchun Street in the north to secondary roads. These areas boast easy accessibility, favorable geographic positioning, and high population densities and foster a "three main and multiple vice" spatial layout. Shopping facilities, on the other hand, are primarily concentrated in the vicinity south of the Northern Expressway, east of Yatai Street. This zone also encompasses adjacent areas surrounding main roads, such as Bei'an Road, Qingming Street, Guilin Road, and other major thoroughfares. Meanwhile, financial and insurance facilities occupy a distinct geographical region, stretching from the east of Xi'an Avenue to the western extent of Changchun Street and south of Xinfa Road. This sector takes on a fan-shaped configuration, encompassing historical sites such as the former location of the Central Bank of the pseudo-Manchu. The influence of historical factors and deep-rooted heritage is palpable, extending to areas around the Yatai Street and Jilin Avenue intersection, the northern section of People's Street, West Democracy Street, Pukyong Road, Minfeng Street, Xinggong Street, and neighboring streets. In summary, the distribution of residential life facilities, catering facilities, and financial and insurance facilities is relatively dispersed and even. Consumers tend to choose these based on proximity. Shopping facilities are relatively concentrated, allowing consumers to select a wide range of goods in a short period. The commercial facilities are generally dense in the west and sparse in the east of the Yitong River, exhibiting a block distribution with clear spatial differences.

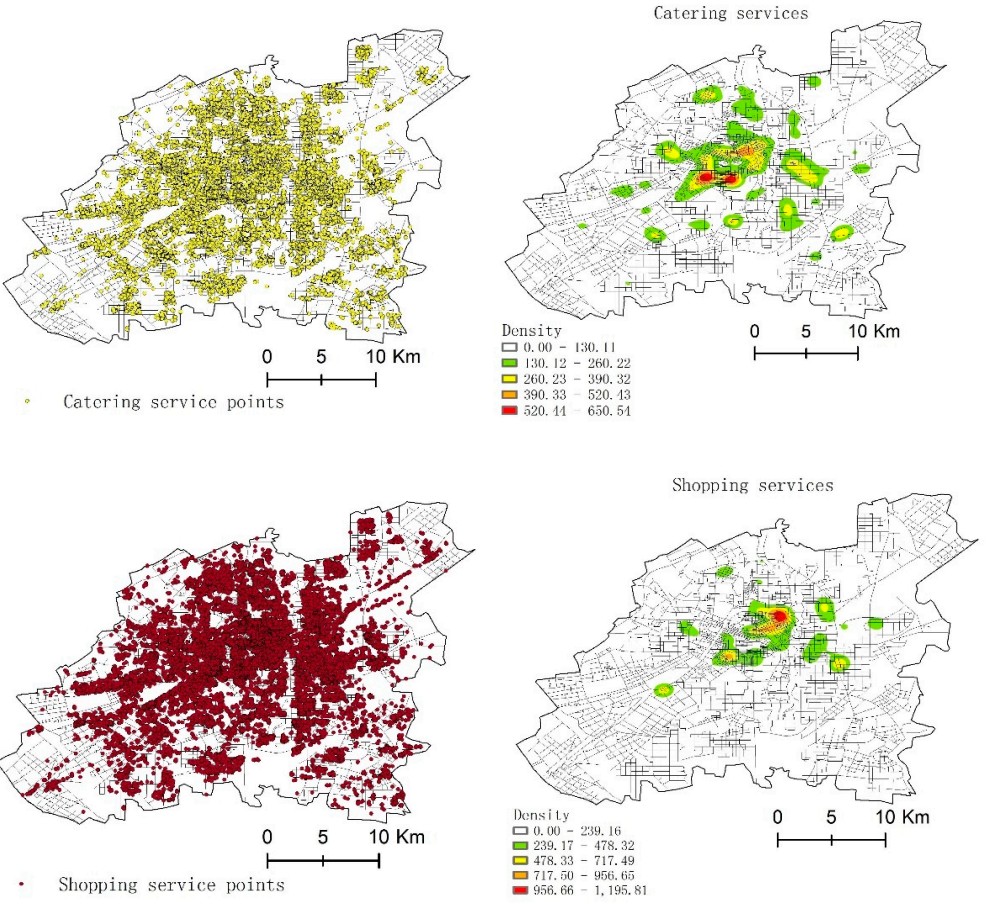

**Figure 3.** *Cont.*

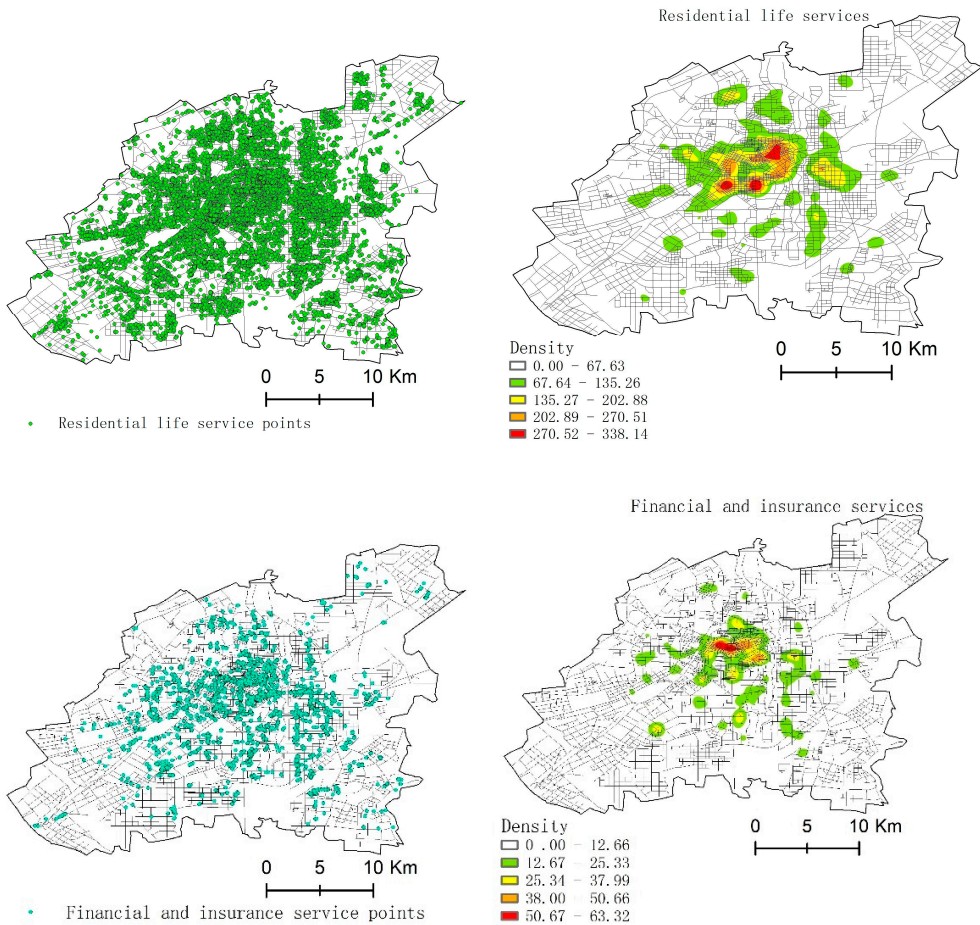

**Figure 3.** Distribution of commercial facility points and their nuclear densities.

## 5. Relationship between Road Network Centrality and Spatial Layout of Commercial Facilities

### 5.1. Centrality Relationship

By utilizing band collection statistics in ArcGIS software, we can directly analyze the spatial correlation between the kernel density of different types of commercial facilities and the centrality kernel density of the transportation network, yielding the direct centrality correlation coefficients of the two, as depicted in Table 2 and Figure 4.

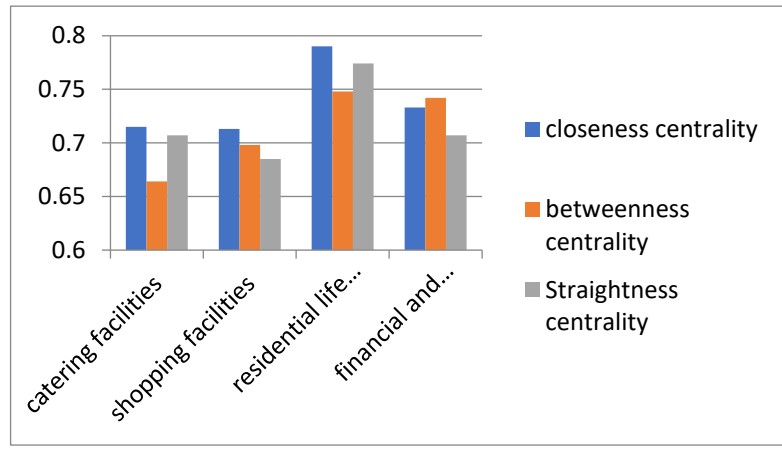

**Figure 4.** Correlation coefficients between commercial facility types and transportation network centrality kernel density.

**Table 2.** Correlation coefficients between commercial facility types and transportation network centrality kernel density.

| Type of Business Service | Catering Facilities | Shopping Facilities | Residential Life Facilities | Financial and Insurance Facilities |
|---|---|---|---|---|
| closeness centrality | 0.715 | 0.713 | 0.790 | 0.733 |
| betweenness centrality | 0.664 | 0.698 | 0.748 | 0.742 |
| straightness centrality | 0.707 | 0.685 | 0.774 | 0.707 |

From Table 2 and Figure 4, it is evident that there is a significant spatial correlation between the road network centrality kernel density and the kernel density of various commercial facility types. Notably, the correlation coefficient between the kernel density of residential life facilities and the closeness centrality kernel density is the highest at 0.790. This indicates that residential life facilities tend to be distributed in areas with high closeness centrality. Additionally, the correlation coefficient between the kernel density of financial and insurance facilities and the closeness centrality kernel density is also high, measuring 0.733. This suggests that financial and insurance facilities tend to be distributed in areas with high closeness centrality, such as near the railway station, People's Square, and Guilin Road (CBD). The average distance from the transportation nodes to all the nodes in these areas is smaller. The correlation coefficients of the residential life facility kernel density and the financial and insurance facility kernel density with the betweenness centrality kernel density are notably high at 0.748 and 0.742, respectively, suggesting that these two facility types are primarily situated in areas characterized by high road network betweenness centrality. Notable locations include the northern section of Renmin Avenue, Changchun Avenue, Chongqing Road, Jiefang Avenue, the intersection of Nanhu Avenue and Renmin Avenue, the intersection of the Southern Expressway and Renmin Avenue, the intersection of East Ring Road and Jilin Expressway, and other nearby areas. These regions witness a higher frequency of the shortest paths, resulting in substantial traffic flow. The correlation coefficients of the residential life facility kernel density, catering facility kernel density, and financial and insurance facility kernel density with straightness centrality kernel density are notably higher at 0.774, 0.707, and 0.707, respectively. This implies that these three types of commercial facility types are predominantly distributed in areas with high straightness centrality and high transportation efficiency. These areas exhibit a scenario where the shortest distance between two nodes in the network closely approximates the straight-line distance between the two nodes. Notable locations include People's Square and the vicinity of the railway station; the area formed by the northern section of People's Street, Changchun Street, Yatai Street, and Zhujiang Road; the area formed by Xinmin Street, Freedom Street, People's Street, and Jiefang Street; and the intersections of Xigangdajie, Beijing Street, and Hankou Street, Xi'an Hutong, Jianshe Street, Xichaoyang Nanhutong, Hwagwang Lu, Hoonchun Street East Hutong, and so on.

## 5.2. Statistical Characteristics of Centrality Relationships

The analysis of the relationship between the urban road network and commercial facilities involved the use of the Pearson correlation coefficient. The centrality of the road network and the kernel density value of commercial facilities were calculated using ArcGIS software. A grid with a specification of 500 m ∗ 500 m was created for systematic grid processing. The 'Extract Multi Values To Points' function in ArcGIS software was then utilized to acquire the analyzed kernel density values within the grid network. Subsequently, the processed data were imported into SPSS 26 software for comprehensive correlation analysis. This analysis entailed calculating the correlation coefficients between road network centrality and various categories of commercial facilities, as presented in statistical Table 3. The correlations calculated using SPSS software are similar to those obtained through band collection statistics, indicating that no errors were introduced by data collection, and there are no inconsistencies observed between the correlation coefficients of commercial facility kernel density and road network centrality kernel density.

**Table 3.** Pearson correlation coefficients between road network centrality and commercial facilities.

| Type of Business Service | Catering Facilities | Shopping Facilities | Residential Life Facilities | Financial and Insurance Facilities |
|---|---|---|---|---|
| closeness centrality | 0.716 ** | 0.718 ** | 0.790 ** | 0.731 ** |
| betweenness centrality | 0.664 ** | 0.698 ** | 0.748 ** | 0.741 ** |
| straightness centrality | 0.708 ** | 0.685 ** | 0.775 ** | 0.704 ** |

** At the 0.01 level (two-tailed), the correlation was significant.

The coefficient of the impact of road network centrality on catering facilities is 0.713 ** (closeness centrality) > 0.708 ** (straightness centrality) > 0.664 ** (betweenness centrality), and the coefficient of the impact of road network centrality on residential life facilities is 0.790 ** (closeness centrality) > 0.775 ** (straightness centrality) > 0.748 ** (betweenness centrality). The spatial layout of catering facilities and residential life facilities is primarily influenced by closeness centrality, followed by straightness centrality, and has the least impact from betweenness centrality. The analysis indicates that catering facilities and residential life facilities exhibit a preference for distribution in areas characterized by high values of closeness centrality. In such areas, network nodes are situated at the shortest average distance from all points in the transportation network, facilitating convenient consumer access to consumption locations via the shortest routes. Nearby areas include Shanghai Road, Gongnong Avenue, Guilin Road, Renmin Street, Bei Hutong of Nanchang Road, Rongguang Road, etc. Catering facilities and residential life facilities typically cater to a large and diverse consumer base characterized by crowd dispersal, frequent consumption, high substitutability of goods, and a priority on convenience over cost-effectiveness. Consumers seek to acquire their desired products in the shortest possible time. Consequently, the spatial layout of catering facilities and residential life facilities takes into account both closeness centrality and straightness centrality in its consideration. The spatial arrangement of catering facilities and residential life facilities is minimally influenced by betweenness centrality. Regions with elevated betweenness centrality values experience a higher volume of shortest paths passing through them, resulting in increased traffic flow. These areas are often associated with urban centers. Rental prices are high in places with good urban transportation and decrease with increasing distance from the city center [34]. Areas with high betweenness centrality tend to command higher land prices, while the spatial arrangement of catering facilities and residential life facilities is less sensitive to intermediation.

The coefficient of the influence of road network centrality on shopping facilities is 0.716 ** (closeness centrality) > 0.698 ** (betweenness centrality) > 0.685 ** (straightness centrality), and the spatial distribution of shopping facilities is primarily influenced by closeness centrality, followed by betweenness centrality, with straightness centrality having the least impact. Upon analysis, it was observed that shopping facilities predominantly cater to urban consumers. The spatial layout is strategically designed to minimize the average distance for consumers to reach shopping destinations, ensuring the shortest possible commuting distances. Notably, areas characterized by high closeness centrality boast the shortest average distances for network nodes to access all points in the transportation network. Consequently, shopping facilities exhibit the strongest correlation with closeness centrality. When consumers have specific requirements for their purchases and are willing to invest more time in comparing products (clothing, household appliances), they tend to gravitate toward areas with better betweenness centrality and higher traffic flow. Additionally, facilities selling products that are bulky or heavy and require a larger display area are often situated in the periphery of shopping facilities centers, taking into account freight costs and the need for efficient transportation routes. In such cases, consumers prioritize reaching their destination in the shortest possible distance, prompting shopping facilities to factor in betweenness centrality in their layout decisions. Meanwhile, straightness centrality, which primarily signifies transportation efficiency, has the least impact on the layout

of shopping facilities when consumers prioritize specific product requirements and are willing to invest extra time in product comparison.

The coefficient of the influence of road network centrality on financial and insurance facilities is 0.741 ** (betweenness centrality) > 0.731 ** (closeness centrality) > 0.704 ** (straightness centrality). The spatial distribution of financial and insurance facilities is primarily influenced by betweenness centrality, followed by closeness centrality, with straightness centrality having the least impact. Financial and insurance facilities are primarily concentrated in regions characterized by higher betweenness centrality, as betweenness centrality serves as a crucial indicator of traffic flow. These areas experience high traffic flow and pass through more shortest paths, such as People's Square, near the railway station, Changchun Street, Xinmin Street, Freedom Avenue, People's Street, and other areas near the main roads. Financial and insurance facilities cater to a larger clientele, making it essential for consumers to access facility points conveniently via the shortest routes. Therefore, the layout of these facilities also considers locations with high closeness centrality. Additionally, when consumers engage in financial and insurance transactions, they often meticulously consider the unique nature of their purchases, with minimal concern for time costs. Consequently, the layout of financial and insurance facilities places less emphasis on the efficiency of transportation access, resulting in the weakest correlation with straightness centrality.

## 6. Comparison of Transportation Network Centrality and Population Distribution Impacts

The influence of the population distribution on the spatial distribution density of different commercial facilities cannot be ignored. Whether the spatial distribution density of commercial facilities is greatly affected by the centrality of the traffic network or by the population distribution needs to be further verified.

Through the utilization of the spatial difference method based on the surface domain, a population density distribution map of the primary urban area of Changchun City was generated (refer to Figure 5). The population distribution exhibits a relatively dispersed pattern, with its central concentration area displaying a comparatively low degree of overlap with the centers of the nuclear density distribution for various types of commercial facilities.

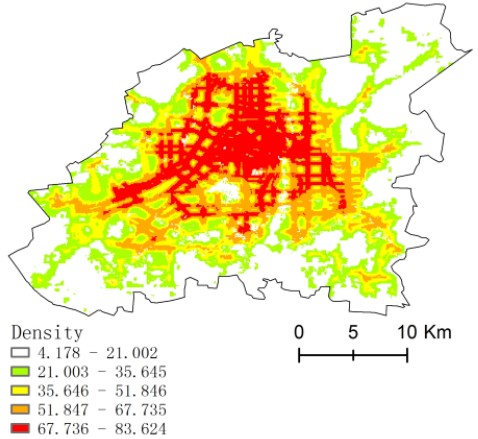

**Figure 5.** Population density distribution map.

Multiple linear regression is a statistical model that establishes the relationship between multiple independent variables and one dependent variable. The interpretability of this model is relatively strong because it can simultaneously consider the impact of multiple independent variables on the dependent variable. Its goal is to identify appropriate regression coefficients that minimize the prediction error of the model for observed data. Using a 500 m by 500 m fishnet grid as the fundamental unit, we extracted population density values from each grid cell and imported them into SPSS software. We then developed univariate regression models to analyze the parameter estimates between the dependent

variable and each independent variable. Subsequently, based on these findings, we constructed multiple regression models correlating population density with the centrality kernel density of the road network and the facility kernel density of various commercial facilities, resulting in 12 models. The analysis reveals that the adjusted R-squared values for all models are above 0.5, indicating that the regression equations exhibit a high degree of fit for the dependent variable. Furthermore, in all 12 models, the independent variables are statistically significant at the 0.05 level. The results are presented in Table 4.

**Table 4.** Regression analysis is performed for different road network centrality cores.

| Model | | B | Standard Error | Beta | t | Sig | $R^2$ | Adjusted $R^2$ |
|---|---|---|---|---|---|---|---|---|
| 1 | constant | −4.480 | 1.897 | | −2.362 | 0.018 | | |
| | population | 0.868 | 0.055 | 0.274 | 15.712 | 0.000 | 0.562 | 0.562 |
| | closeness centrality | 3.211 | 0.100 | 0.557 | 32.021 | 0.000 | | |
| 2 | constant | −2.475 | 0.211 | | −1.231 | 0.219 | | |
| | population | 1.043 | 0.057 | 0.329 | 18.303 | 0.000 | 0.516 | 0.615 |
| | betweenness centrality | 0.000 | 0.000 | 0.481 | 26.811 | 0.000 | | |
| 3 | constant | −11.306 | 1.912 | | −5.912 | 0.000 | | |
| | population | 0.862 | 0.057 | 0.272 | 15.163 | 0.000 | 0.548 | 0.548 |
| | straightness centrality | 0.000 | 0.000 | 0.546 | 30.466 | 0.000 | | |

Dependent variable: catering facilities.

| Model | | B | Standard Error | Beta | t | Sig | $R^2$ | Adjusted $R^2$ |
|---|---|---|---|---|---|---|---|---|
| 4 | constant | −4.834 | 2.674 | | −1.808 | 0.071 | | |
| | population | 0.809 | 0.078 | 0.187 | 10.394 | 0.000 | 0.580 | 0.581 |
| | closeness centrality | 4.753 | 0.141 | 0.605 | 33.621 | 0.000 | | |
| 5 | constant | −0.392 | 2.732 | | −0.144 | 0.886 | | |
| | population | 0.935 | 0.077 | 0.216 | 12.087 | 0.000 | 0.556 | 0.556 |
| | betweenness centrality | 0.000 | 0.000 | 0.578 | 32.288 | 0.000 | | |
| 6 | constant | −14.978 | 2.757 | | −5.433 | 0.000 | | |
| | population | 0.872 | 0.082 | 0.201 | 10.635 | 0.000 | 0.519 | 0.520 |
| | straightness centrality | 0.000 | 0.000 | 0.565 | 29.824 | 0.000 | | |

Dependent variable: shopping facilities.

| Model | | B | Standard Error | Beta | t | Sig | $R^2$ | Adjusted $R^2$ |
|---|---|---|---|---|---|---|---|---|
| 7 | constant | −1.309 | 0.981 | | −1.335 | 0.182 | | |
| | population | 0.481 | 0.029 | 0.256 | 16.868 | 0.000 | 0.668 | 0.668 |
| | closeness centrality | 2.198 | 0.052 | 0.642 | 42.387 | 0.000 | | |
| 8 | constant | 0.333 | 1.052 | | 0.317 | 0.751 | | |
| | population | 0.577 | 0.030 | 0.306 | 19.374 | 0.000 | 0.625 | 0.624 |
| | betweenness centrality | 0.000 | 0.000 | 0.578 | 36.566 | 0.000 | | |
| 9 | constant | −5.985 | 1.009 | | −5.932 | 0.000 | | |
| | population | 0.486 | 0.030 | 0.258 | 16.221 | 0.000 | 0.644 | 0.643 |
| | straightness centrality | 0.000 | 0.000 | 0.621 | 39.026 | 0.000 | | |

Dependent variable: residential life facilities.

| Model | | B | Standard Error | Beta | t | Sig | $R^2$ | Adjusted $R^2$ |
|---|---|---|---|---|---|---|---|---|
| 10 | constant | −0.874 | 0.161 | | −5.418 | 0.000 | | |
| | population | 0.052 | 0.005 | 0.194 | 11.104 | 0.000 | 0.560 | 0.560 |
| | closeness centrality | 0.302 | 0.009 | 0.619 | 35.491 | 0.000 | | |
| 11 | constant | −0.530 | 0.159 | | −3.326 | 0.001 | | |
| | population | 0.577 | 0.005 | 0.203 | 12.115 | 0.000 | 0.578 | 0.577 |
| | betweenness centrality | 0.000 | 0.000 | 0.628 | 37.463 | 0.000 | | |
| 12 | constant | −1.520 | 0.167 | | −9.122 | 0.000 | | |
| | population | 0.055 | 0.005 | 0.207 | 11.237 | 0.000 | 0.524 | 0.523 |
| | straightness centrality | 0.000 | 0.000 | 0.581 | 31.582 | 0.000 | | |

Dependent variable: financial and insurance facilities.

In summary, the analysis reveals a significant positive influence of road network centrality and population distribution on the distribution of various types of commercial facilities. More specifically, it is evident that the impact of the population kernel density on various types of commercial facilities is considerably less pronounced compared to the influence of the road network centrality kernel density. This substantiates that road network centrality has a greater influence on the distribution of commercial facility types than population distribution.

## 7. Discussion and Conclusions

### 7.1. Discussion

#### 7.1.1. Improving the Layout of Transport Networks

Transportation plays a pivotal role in shaping human–environment systems, with urban transport research occupying a central position in geographic studies [36]. The spatial configuration of transportation networks inherently mirrors the dynamic interconnections within a city. Urban transport networks embody spatial, social, and economic dimensions, serving as a comprehensive indicator for location evaluation. Beyond reflecting the spatial characteristics of a city, they also offer insights into the lifestyle preferences and socioeconomic characteristics of its residents. In modern society, many functional aspects rely on the road network [37]. By unveiling the urban road network, we can gain a better understanding of the city's evolutionary process [38]. Standardizing urban road networks while preserving the ecological environment is essential. Enhancing the layout of transport networks in central urban areas involves strengthening the connectivity of expressways and main roads to surrounding areas. This improvement aims to enhance service levels and travel experiences. By achieving a more balanced distribution of channels and smoother travel, urban areas can experience an overall enhancement in transport network centrality. Opening up "dead-end roads" can indeed improve traffic efficiency and expand the coverage of urban road networks. Furthermore, aligning road planning and construction with the urban functional layout is crucial for long-term development. Optimizing the road network layout to create an interconnected system will address the evolving needs of the city and contribute to its sustainable growth. This approach ensures that urban road infrastructure keeps pace with the changing demands of urbanization and facilitates the efficient movement of people and goods within the city.

#### 7.1.2. Improve the Service Layout of Commercial Facilities

Urban infrastructure encompasses a wide range of facilities crucial for the functioning and development of cities. Commercial service facilities play a particularly vital role within this infrastructure, serving as the physical spaces where commercial and service activities take place. These facilities provide essential services to residents and businesses, contributing to the overall economic vitality and quality of life in urban areas. As such, they are integral components of the urban fabric, supporting the diverse needs of urban populations and facilitating economic growth and social interaction. Commercial service facilities are indeed crucial for urban development and the well-being of residents. They not only provide essential goods and services but also contribute to the overall economic prosperity of urban areas. By offering a wide range of services such as retail, dining, entertainment, healthcare, and financial services, these facilities meet the diverse needs of residents and enhance their quality of life. Additionally, commercial service facilities create employment opportunities, attract investment, and stimulate economic activity, thereby contributing to urban vitality and growth. Overall, their role in urban development cannot be underestimated, as they are essential for ensuring the sustainability and prosperity of cities.

The uneven distribution of commercial facilities indeed poses challenges to achieving social equity and ensuring access to essential amenities for all residents. The block-like concentration and significant central aggregation of commercial facilities often result in disparities in access, with certain areas enjoying better access to services while others are

underserved. This spatial inequality can exacerbate social inequalities, as residents in underserved areas may face difficulties in accessing necessary goods and services, leading to disparities in quality of life and opportunities. Addressing these spatial imbalances requires targeted interventions. By promoting more balanced distribution and improving accessibility, cities can work toward creating more inclusive and equitable urban environments for all residents. The block-like clustering of commercial service facilities along commercial streets and major transportation routes indeed highlights the influence of traffic patterns and urban planning on their distribution. However, the imbalance in the distribution of different types of commercial service facilities suggests that certain areas may be underserved or lack access to specific types of services. To address this imbalance, urban planners should consider strategies to encourage the clustering of commercial facilities along bypasses and secondary trunk roads within the urban road network. By strategically locating these facilities, cities can improve accessibility and ensure that residents have equitable access to a diverse range of essential services, thereby promoting a more balanced and inclusive urban environment. Strategically arranging catering facilities and life service facilities around the intersections of bypasses and secondary trunk roads can indeed enhance accessibility and convenience for residents. Additionally, in areas with higher proximity, commercial service facilities with high passenger flow demands are strategically distributed to align with market demand. Exploring the underlying patterns of residents' demand for each industry and market principles can provide valuable insights for future research and urban planning efforts. By understanding these patterns, urban planners can better anticipate the needs of residents and tailor the distribution of commercial service facilities to meet those needs effectively, ultimately contributing to a more vibrant and sustainable urban environment.

### 7.1.3. Significance

Changchun is a typical single-core inland city, where the traditional grid form constitutes the basic road network structure of its main urban area, providing a framework for the city's traffic. The design of roundabout plazas facilitates traffic flow. Our focus is on the main urban area of Changchun, rather than the broader metropolitan area. We aim to study and analyze the relationship between the centrality of the road network and the spatial distribution of commercial facilities through more refined solutions, exploring the impact mechanisms of their distribution patterns. By doing so, we can accurately grasp the influence of road network centrality in the main urban area of Changchun on the spatial distribution of commercial facilities. By conducting an analysis and drawing insights from existing research on road network centrality and commercial facilities, we observe the substantial impact of road network centrality on the layout of commercial facilities. This underscores the potential for optimization by reconfiguring major urban roads in Changchun, thereby enhancing the road network's capacity to support commercial facilities. These efforts can lay the groundwork for a new urban commercial facility pattern, contributing significantly to urban development.

### 7.1.4. Shortcomings

Traffic flow, pedestrian movement, and other factors significantly impact the transportation network of a region. This study specifically focused on the centrality characteristics of the road network, examining the influence of road network centrality within the study area on the distribution of commercial facilities. Public transportation routes also affect the quality of the transportation network [11,12] but were not considered in this analysis due to data acquisition limitations. Future research, building on this foundation, will incorporate additional factors, such as public transit routes, passenger traffic volume, modes of travel, and time costs, to provide a more comprehensive analysis of the impact of the road network on the distribution of commercial facilities. This study is confined to a case analysis of Changchun City, which inherently possesses certain limitations. To further

substantiate the findings of this research, it is imperative to conduct analyses of additional cases, contingent upon the availability of relevant data.

*7.2. Conclusions*

In this study, we examined the interplay between road network centrality and the distribution of various categories of commercial facilities, with a specific emphasis on spatial clustering. Through the application of methods such as kernel density estimation and Pearson's correlation coefficient, we delved into the intricate relationship between road network centrality and the spatial arrangement of commercial facilities. The key findings of this study are as follows:

(1) Closeness centrality exhibits a "Core–Periphery" pattern, and both closeness centrality and straightness centrality reveal a polycentric structure within the core density distribution map. In contrast, betweenness centrality predominantly concentrates in the vicinity of the city's main road network. The spatial layout of commercial facilities exhibits significant disparities, influenced by urban planning. The railway and bus stations have a strong radiating effect to the west, while development toward the west remains weak. Additionally, several large commercial hubs (Chongqing Road, Hongqi Street, and Guilin Road) are located to the west of the Yitong River. This results in a distribution of commercial facilities demarcated by the Yitong River, with a dense concentration in the west and a sparse arrangement in the east, presenting a block-like pattern.

(2) A significant correlation exists between the spatial distribution of commercial facilities and road network centrality. Notably, various types of commercial facilities are influenced by road network centrality in distinct ways. From a centrality perspective, residential life facilities are most strongly affected by road network centrality; residential life facilities, including public toilets, logistics centers, beauty salons, and business halls, are distributed throughout the city to ensure residents have easy access wherever there is a population. Following these are financial and insurance facilities, which are often located near People's Square, Guilin Road, and other hubs of economic activity, providing convenient access to these services. Catering facilities follow in significance, with shopping facilities being the least extensive. Shopping facilities, which mainly include markets and supermarkets, consider several factors in their distribution. On the one hand, the distance for consumers is considered to ensure convenience and reduce the cost of goods distribution. On the other hand, the size of the area occupied by these facilities is also a factor. Consequently, shopping facilities are least affected by centrality.

(3) Road network centrality plays a pivotal role in determining the placement of commercial facility points. Notably, there are variations in the extent of its influence on different types of commercial facilities, resulting in directional tendencies in the layout of commercial facilities. The distribution of amenity, life, catering, and shopping facilities is primarily influenced by closeness centrality, where road network nodes in the area boast the shortest average distances to reach all other road network nodes. In contrast, betweenness centrality exerts a significant impact on the selection of locations for financial and insurance facilities and residential life facilities. This effect is attributed to the higher intermediation levels within the road network in these regions, resulting in increased traffic flow through the shortest paths and elevated traffic volumes. Furthermore, the spatial layout of the road network contributes to the phenomenon where the shortest distance between two nodes in the road network closely approximates the straight-line distance between those nodes. Consequently, areas with better straightness centrality are favored for the placement of residential life facilities, financial and insurance facilities, and catering facilities. Straightness centrality serves as a crucial index indicative of transportation efficiency, thereby influencing the selection of locations that offer better commuting convenience.

(4) In areas with a dense distribution of commercial facilities, the population is also relatively concentrated, exemplified by the universities near Guilin Road. Through comparisons of road network centrality, it has been observed that population distribution has a minimal impact on the layout of different commercial facilities.



This study focused on exploring the correlation between urban road network centrality and the spatial distribution of commercial facilities. To validate the generalizability of our findings, additional case studies are recommended. Future research endeavors should extend the road network centrality framework by incorporating control variables. This approach will facilitate a more comprehensive understanding of the factors influencing the distribution of commercial facilities, ultimately contributing to the improvement of urban spatial structures and the promotion of healthy city development. This research provides a robust theoretical foundation for future investigations.

**Author Contributions:** Conceptualization, X.S., D.L. and J.G.; methodology, X.S. and D.L.; software, X.S. and D.L.; validation, X.S., D.L. and J.G.; formal analysis, J.G.; investigation, X.S. and D.L.; resources, X.S. and D.L.; data curation, X.S. and D.L.; writing—original draft preparation, X.S. and D.L.; writing—review and editing, D.L.; visualization, D.L.; supervision, J.G.; project administration, X.S. and D.L.; funding acquisition, D.L. All authors have read and agreed to the published version of the manuscript.

**Funding:** This research was funded by the National Natural Science Foundation of China (42171236); Young Scientist Group Project of Northeast Institute of Geography and Agroecology, Chinese Academy of Sciences (2022QNXZ02).

**Institutional Review Board Statement:** Not applicable.

**Informed Consent Statement:** Not applicable.

**Data Availability Statement:** Data are contained within the article.

**Conflicts of Interest:** The authors declare no conflict of interest.

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
