# Peer review of "A Study on the Relationship between Road Network Centrality and the Spatial Distribution of Commercial Facilities—A Case of Changchun, China"

_sustainability, doi:10.3390/su16103920_

Round 1

Reviewer 1 Report

Comments and Suggestions for Authors

The paper is very well written and is a compelling analysis on the relationship between road network centrality and the spatial distribution of commercial facilities in Changchun City (China). I will not list all the merits of the paper - there are indeed plenty of them, it is a good and thorough research. I would only point out the weaknesses of the paper. Apart from some minor mistakes in terms of English language (more detailed below, in the specific section), the main problem of this paper is that it addresses only Changchun City in China and nothing else. It is not really a case study, it is a full study on this specific city, and I agree that this study may be of great importance for the local authorities in Changchun City, but it does not have a larger significance, it does not have any meaning at a national, continental or global level, nothing can be truly inferred out of it, authorities or scientists in other parts of the world cannot gain anything out of it, because it cannot be generalized. Another issue is that it is too long and sometimes there are too many descriptions, which are not entirely necessary. For those who do not know Changchun City, these descriptions serve for nothing. On the other hand, figures are too small and it is difficult to see anything on them, not mentioning their legend, which is almost illegible.

Comments on the Quality of English Language

The quality of English language is generally good, but there are some mistakes which must be corrected. Even the title is wrong, it should be ”distribution” (and not ”distribition”) and ”facilities” (in plural form, not ”facility”). Then, the authors should decide whether it is a comma or a full stop (row 56), there is a word missing in the sentence ”on analyzing and economic activities” or perhaps ”and” should be removed (row 61), the word ”services” appears twice in red (rows 295 and 297) without any reason, the word ”facility” is often used in singular form, when it should be plural, in ”commercial facilities” (for instance, rows 497, 499, 511, 526, 529, 531, 532, 533, 535, 539, 542, 551, 552, and so on).

Reviewer 2 Report

Comments and Suggestions for Authors

This paper focuses on exploring the correlation between urban road network centrality and the spatial distribution of commercial facility in the main urban 11 area of Changchun.The author provides relevant research results. However, there are some problems in the paper that need to be revised, and these issues are as follows:

1)Roadway network data encompassing all types of roads within the City of Changchun in the year 2022 was collected。Where these data come from should be explained.

2)The map of China in Figure 1 should use the Chinese standard map and indicate where to download it.

3)At the beginning of the method description section, the research method and the problems that need to be solved of the entire paper should be explained, but the author only explained some quoted formulas.

4)Figures 2 and 3 are unclear and cannot be used to illustrate the problem.

5)The conclusion in section 7.2 is too abstract and does not take into account the local geographical characteristics. It is recommended to supplement the geographic content.

Reviewer 3 Report

Comments and Suggestions for Authors

This manuscript presents a method for evaluating transportation networks based on their centrality in the main urban area of Changchun. The authors miss some major points that are not relevant to making their work publishable in such a reputable journal. First, the language usage is not appropriate and needs to be rewritten for major parts of the manuscript, including the abstract. Second, the literature review is very weak and needs more focus. The literature has no clue about other network evaluation methods to be compared with the selected one. Also, the results of these studies regarding their cases of studies should be discussed within this manuscript and compared with its results. See for examples, the evaluation of transit networks based on its transfer number in Design scheme of multiple-subway lines for minimizing passengers transfers in mega-cities transit networks and An optimal metro design for transit networks in existing square cities based on non-demand criterion. Third, there is no justification for the use of Kernel Density Estimation (KDE) over other measures or even a comparison. Also, the use of Multiple linear regression is vague without a real discussion of the assumed linearity. I did not see the goodness of fit measurements like coefficient of determination (R2) in the results. I believe that machine learning tools would be more appropriate. Finally, Figures like Figure 3 need to be added with higher resolution, and Figure 4 needs to be redrawn in a more scientific (i.e., journal publishable) way.

Comments on the Quality of English Language

The language usage is not appropriate and needs to be rewritten for major parts of the manuscript, including the abstract

Round 2

Reviewer 3 Report

Comments and Suggestions for Authors

The authors have addressed all my concerns adequately.

Comments on the Quality of English Language

None.